# Breaking solvation dominance of ethylene carbonate via molecular charge engineering enables lower temperature battery

Yuqing Chen [1], Qiu He[2], Yun Zhao [3], Wang Zhou[1], Peitao Xiao [4], Peng Gao[1], Naser Tavajohi [5], Jian Tu[6], Baohua Li [3], Xiangming He [7], Lidan Xing [8], Xiulin Fan [9] & Jilei Liu [1] ✉

Low temperatures severely impair the performance of lithium-ion batteries, which demand powerful electrolytes with wide liquidity ranges, facilitated ion diffusion, and lower desolvation energy. The keys lie in establishing mild interactions between Li⁺ and solvent molecules internally, which are hard to achieve in commercial ethylene-carbonate based electrolytes. Herein, we tailor the solvation structure with low-ε solvent-dominated coordination, and unlock ethylene-carbonate via electronegativity regulation of carbonyl oxygen. The modified electrolyte exhibits high ion conductivity (1.46 mS·cm⁻¹) at −90 °C, and remains liquid at −110 °C. Consequently, 4.5 V graphite-based pouch cells achieve ~98% capacity over 200 cycles at −10 °C without lithium dendrite. These cells also retain ~60% of their room-temperature discharge capacity at −70 °C, and miraculously retain discharge functionality even at ~−100 °C after being fully charged at 25 °C. This strategy of disrupting solvation dominance of ethylene-carbonate through molecular charge engineering, opens new avenues for advanced electrolyte design.

Lithium-ion batteries (LIBs) that can operate over a wide temperature range at high voltages, are highly sought-after for energy storage under extreme conditions such as polar expeditions, high-altitude stations, and certain military applications. However, several challenges remain, including: (i) the narrow liquid ranges of electrolytes[1], (ii) slow mass transport, giving poor conductivity and Li diffusivity in bulk materials[2,3], especially graphite anodes, and (iii) sluggish charge transfer processes resulting from high energy barriers for Li⁺ desolvation and Li⁺ migration within the solid electrolyte interface (SEI)[2,4–9].

These factors lead to unwanted lithium plating or unfriendly SEIs, resulting in low efficiencies and serious safety concerns[10–15]. (Supplementary Fig. 1a). Previous studies have shown that charge transfer processes rather than mass transport dominate the electrochemical capabilities of LIBs at low temperatures (LT) (≤0 °C), given the much higher energy barrier for the former[1,8,9,16,17]. This highlights the importance of reducing the desolvation energy barriers to facilitate charge transfer[4,6,16]. The desolvation energy is highly dependent on the solvation structure of the electrolyte[9,16,18,19]. Therefore, the keys to a

[1]College of Materials Science and Engineering, Hunan Joint International Laboratory of Advanced Materials and Technology for Clean Energy, Hunan Province Key Laboratory for Advanced Carbon Materials and Applied Technology, Hunan University, Changsha 410082, People's Republic of China. [2]College of Materials Science and Engineering, Sichuan University, Chengdu 610065, P. R. China. [3]Institute of Materials Research, Tsinghua Shenzhen International Graduate School, Tsinghua University, Shenzhen 518055, China. [4]College of Aerospace Science and Engineering, National University of Defense Technology, Changsha 410073, China. [5]Department of Chemistry, Umeå University, Umeå 90187, Sweden. [6]LI-FUN Technology Corporation Limited, Zhuzhou 412000 Hunan, China. [7]Institute of Nuclear and New Energy Technology, Tsinghua University, Beijing 100084, China. [8]Engineering Research Center of MTEES (Ministry of Education), Research Center of BMET (Guangdong Province), Engineering Lab. of OFMHEB (Guangdong Province), Key Lab. of ETESPG (GHEI), And Innovative Platform for ITBMD (Guangzhou Municipality), School of Chemistry, South China Normal University, Guangzhou 510006, China. [9]State Key Laboratory of Silicon Materials, School of Materials Science and Engineering, Zhejiang University, Hangzhou 310027, China. ✉e-mail: liujilei@hnu.edu.cn

moderate desolation process lie in constructing a Li⁺ solvation shell with weak interactions between Li⁺ and the solvents[9].

Strategies including the use of liquefied gas electrolytes[20,21], novel cosolvents[22,23], highly fluorinated solvents[4,24], PC-based electrolytes[25], local high-concentration electrolytes (LHCEs)[4,8,23,26,27], weakly solvating electrolytes[4,8,9,23,25,27–29], and cointercalation methods[30] have therefore been proposed to optimize solvation structures. These strategies are designed to increase the proportion of anions in the Li⁺ solvation shell, which is mainly achieved by replacing ethylene carbonate (EC) with low dielectric constant (ε) solvents. Among these strategies, LHCEs are particularly promising for LT battery applications. These systems exploit the low polarities of fluorinated ether diluents to break the strong interactions between highly polar (high-ε) molecules in the electrolyte, which broadens the liquid range of the electrolyte and facilitates desolvation (Supplementary Fig. 1b). However, these EC-free strategies lower the desolvation energy at the expense of conductivity, and the direct interactions between high-ε solvents and Li⁺ are ignored. This impedes potential applications in extreme low-temperature environments (i.e., −60 °C or much lower).

In LIB electrolytes, the Li⁺ ions are typically coordinated with polar solvent molecules in carbonate electrolytes via the electronegative carbonyl oxygens. Theoretically, one way to weaken this coordination without sacrificing the high-ε property of the polar solvent is to reduce the electronegativity of the carbonyl oxygen in the high-ε solvent, rather than replacing it with low-ε solvents. This could potentially be achieved by introducing strongly electron-withdrawing elements[16,24], such as fluorine, for example. Fluorination is expected to weaken the coordinating interactions between high-ε solvents (i.e., cyclic carbonates) and Li⁺ (Fig. 1a), resulting in (i) the release of more high-ε solvent, which would facilitate coordination of the low-ε solvents (i.e., linear carbonates) with Li⁺ to occupy its coordination sites, thus converting the solvent separated ion pairs (SSIPs) from high-ε solvent dominated to low-ε dominated, and (ii) unlocked interaction of the remaining coordinated high-ε solvents (Fig. 1a). The unique solvation structure has several merits: (i) the overall coordinating interactions between Li⁺ and the solvents are significantly and thoroughly weakened by reducing both the coordination number and strengths of the high-ε solvents and by promoting the formation of low-ε solvent-dominated solvation structures (Fig. 1a). Moreover, the fluorinated carboxylate cosolvent would also participate in solvation via a carbonyl group, even though its interaction with Li⁺ is quite weak (Fig. 1a). All of these changes together promote the formation of a desirable solvation structure with much weaker interactions between Li⁺ and all the solvents, which is highly desirable for widening the liquid range and

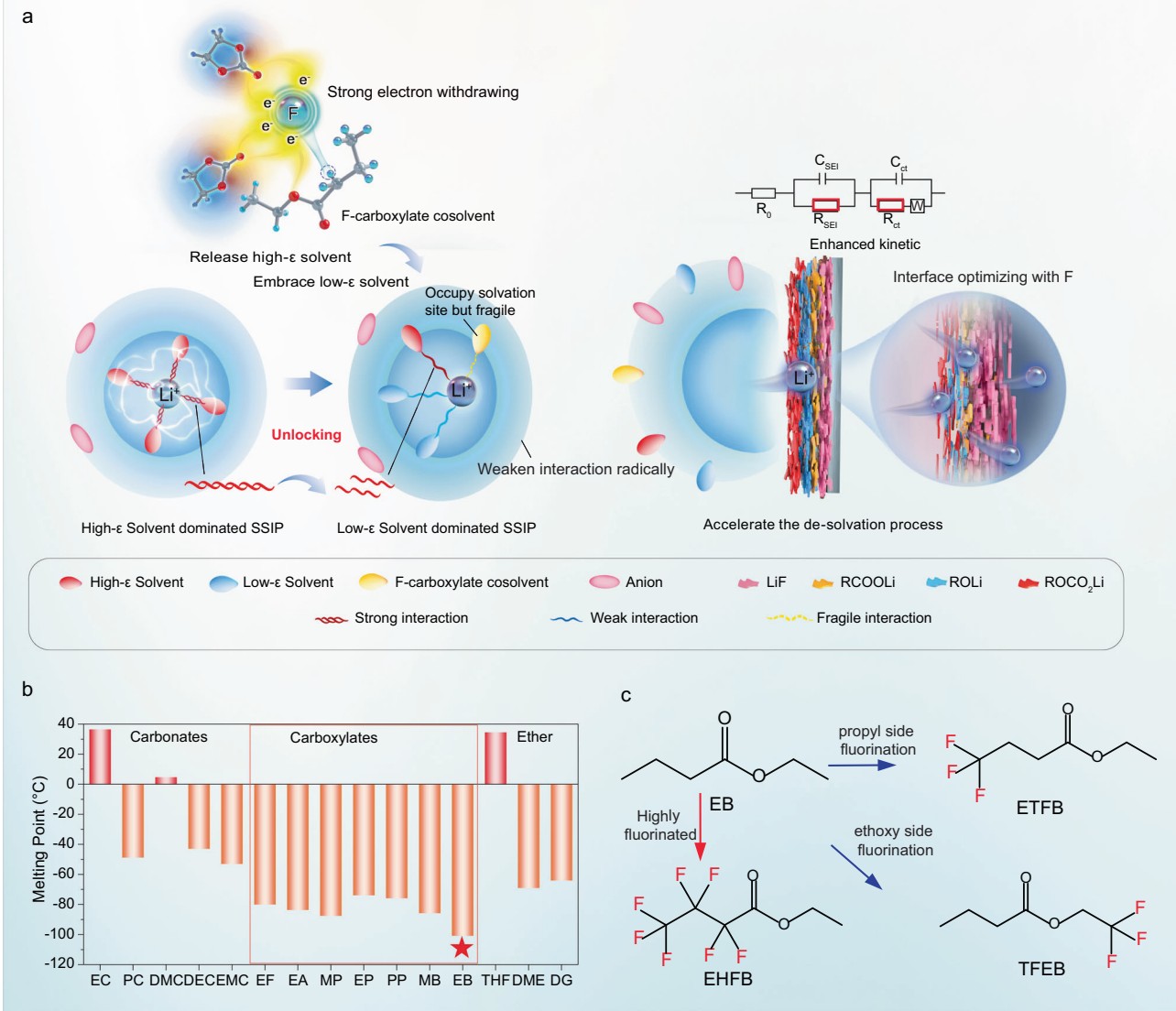

**Fig. 1 | Design strategy and screening principles for potential cosolvents. a** Proposed low temperature electrolyte design principle. **b** Melting point of common carbonate and carboxylate solvents. **c** Chemical structure of the selected EB and its fluorinated analogue cosolvents.

further facilitating Li+ desolvation. (ii) the fluorinated cosolvent would benefit F-rich SEI formation due to the lower LUMO energy[31] (Supplementary Table 2). Additionally, fluorination would lower the electrolyte's HOMO energy, thereby widening its electrochemical window for use in high-voltage batteries[31,32]. This approach offers a more effective way to rejuvenate the desire solvation structure and is expected to extend its application to extremely low temperatures.

Based on these considerations, we identified ethyl butyrate (EB) as a solvent to be fluorinated because of its low melting point (Fig. 1b and Supplementary Fig. 2). A series of high-performance LIB electrolytes containing analogues of EB as cosolvents, including 4,4,4-ethyl trifluorobutyrate (ETFB), ethyl heptafluorobutyrate (EHFB) and 2,2,2-trifluoroethyl butyrate (TFEB), with varying fluorination degrees and/or fluoride sites (Fig. 1c and Supplementary Fig. 3), were developed. As expected, the cosolvent containing electrolyte exhibited high ionic conductivity (1.4-14.54 mS·cm⁻¹) over a wide temperature range (−90 to +70 °C) owing to the formation of a solvation structure dominated by DEC that weakened the overall coordination interactions. Further optimization was achieved by tailoring the fluorination degree and fluoride sites of the cosolvents. Specifically, a higher fluorination degree and/or ethoxy side fluorination resulted in a weaker interaction between Li+ and the solvents and thus better low-temperature performance. Consequently, 4.5 V graphite-based pouch cells (1 Ah) performed stably over 200 cycles at −10 °C with only 2% capacity loss and minimal lithium dendrite formation, and they retained a capacity of 334 mAh even during charge and discharge at −60 °C for one cycle. At the low temperature of −70 °C, these pouch cells utilizing the EHFB electrolyte retained a capacity of 716 mAh under a room temperature charge−low temperature discharge protocol, corresponding to 61% of their room temperature capacity. Furthermore, these cells remained discharge functional even at −100 °C (Supplementary Movie S1, Movie S3, and Supplementary Fig. 40, Supplementary Fig. 49–52). after being fully charged at room temperature, demonstrating the great practical feasibility of the electrolyte design reported here.

## Results

### Physical properties at extreme low temperature

ETFB, EHFB, and TFEB are all fluorinated derivatives of EB (Supplementary Fig. 3a). EB is an ideal cosolvent for low-temperature LIB electrolytes because of its low melting point (−−100.8 °C), and long-alkyl chain which are enable the formation of high-quality SEIs[33]. Fluorinated EB derivatives have lower electron densities around the carbonyl O atoms than either EC or EB (Fig. 2a, Supplementary Table 1), leading to weaker binding with Li+ (Fig. 2b and Supplementary Fig. 3b–4b). Moreover, the degree of fluorination affects the electronegativities of the cosolvents and the Li+-solvent coordination strength (Fig. 2b). For example, EHFB-Li+ had the lowest binding energy (-1.72 eV) among the fluorinated cosolvents, and EC-Li+ in the EHFB system had the lowest binding energy (0.71 eV) among the mixed solvents. These results demonstrated that fluorination not only reduced the coordination of ethyl butyrate itself but also weakened the interactions between the surrounding EC and Li+, and the latter can be explained by the "dipole-dipole effect"[34,35] (Supplementary Fig. 47). Specifically, the strong electron affinity of fluorine led to a shift in the charge distribution from C=O end to -CF₃ end, which changed the dipole-dipole interaction between the fluorinated ethyl butyrate and the surrounding EC molecule and eventually resulted in a weak solvation of Li+ by the EC.

The weak solvation structure significantly improved the physical properties of the electrolyte (Fig. 2f), such as the ion conductivity and liquidity. This occurred because (i) some of the EC molecules were released from the solvation structure, and (ii) more DEC and cosolvent with much lower melting points were involved, resulting in a lower melting point (Fig. 2d) and lower viscosity of the bulk electrolyte (inset of Fig. 2c). Specifically, the estimated freezing points were increased in the order of EHFB (−135 °C), TFEB (−132 °C), and ETFB (−130 °C)

(Fig. 2d), following the same trend as that of the EC-Li+ binding energy. This was supported by the fact that the EHFB electrolyte remained liquid even after 30 min in a −110 °C bath. (Fig. 2e and Supplementary Fig. 5). Based on the viscosity effect and dielectric effect[2] (Fig. 2f), the high ion conductivity depended on both the high permittivity and low viscosity, which were both exhibited by the EHFB electrolyte, with a record-high ionic conductivity of -1.46 mS·cm⁻¹ even at −90 °C (Fig. 2c and Supplementary Table 3)[4]. This demonstrated its great feasibility for use in extremely cold environments. The temperature-dependence of the conductivity was fitted with the Vogel-Fulcher-Tammann (VFT) equation (Supplementary Fig. 6), and the calculated activation energy for the EHFB electrolyte was 0.98 eV, much smaller than those of the other four electrolytes (19.19 eV, 2.98 eV, 1.40 eV and 1.41 eV for base, EB, ETFB and TFEB, respectively), indicating the superior performance of EHFB. In contrast, the base electrolyte froze at -−50 °C (Fig. 2d) and showed poor ionic conductivity when the temperature was decreased to −90 °C (0.001 mS·cm⁻¹) (Fig. 2c), highlighting the benefits of the modified solvation structure.

### Unlocking the electrolyte solvation interaction

The solvation structures were decoupled by molecular dynamics (MD) simulations (Supplementary Fig.7), and the calculated radial distribution functions (RDFs) (g(r)) and coordination numbers (n(r)) for Li+ in each electrolyte case are provided (Fig. 3a, b and Supplementary Figs. 8–9). The first solvation radius for each solvent was -2.65 Å, and $PF_6^-$ occupied the second solvation shell (-4.2 Å) to form separated ion pairs (SIP- $PF_6^-$). In the base electrolyte, EC was the dominant solvent in the first Li+ solvation shell, with a coordination number of 1.31 at room temperature (RT, 25 °C), followed by linear carbonate DEC (1.13) and EMC (0.94) (Fig. 3g and Supplementary Fig.8, Supplementary Table 6). However, the Li+ solvation structure was changed significantly with the addition of the abovementioned cosolvents. Specifically, the EC coordination number dropped sharply to 0.61 (Fig. 3b and Fig. 3g) in the EHFB electrolyte, while the DEC coordination number rose to 1.46 (Fig. 3b and Fig. 3g). Similar trends were observed in the EB, ETFB, and TFEB systems, indicating transitions from an EC-dominant solvation structures to the DEC-dominant solvation structures. These transitions were found to be fluorination dependent, and the higher the fluorination degree of the cosolvent was, the stronger the coordination of the DEC solvents with Li+ and the weaker the interaction between the EC and Li+. These effects were more evident at lower temperatures. For instance, the DEC coordination number in the EHFB electrolyte increased from 1.46 at 25 °C to 1.71 at −70 °C, while the EC coordination number fell from 0.61 to 0.33 (Fig. 3h, Supplementary Fig.9, Supplementary Table 6), indicating that more EC was replaced by DEC in the first solvation shell at LT. In contrast, in the base electrolyte, more EC was coordinated with Li+ upon decreasing the temperature, as identified by the increased coordination number for EC (from 1.34 at 25 °C to 1.76 at −70 °C) and the decreased DEC coordination number (from 1.13 at 25 °C to 0.74 at −70 °C). The decrease in EC coordination inevitably led to a corresponding increase in DEC coordination considering a constant total number of solvent molecules coordinating each Li+. Clearly, the advantage of reducing EC coordination will not be overshadowed by an increase in DEC coordination, which is attributed to the intrinsically low polarity nature of DEC with respect to EC. On one hand, since EC is a high polar solvent, thus a decrease in the coordination number of EC no doubt results in a weakening of solvation. On the other hand, an increase in the coordination number of DEC also leads to a weakening of solvation given its low polarity. Therefore, these two factors synergistically promote the formation of an overall weaken solvation structure. This highlights the merits of the EHFB cosolvent for LT applications. Apart from the effects of fluorinated cosolvents on the solvation of the main solvents, they also affected the Li+ solvation, given that the coordination numbers of EB, ETFB, TFEB, and EHFB decrease from 1.3, 0.85, 0.74, to 0.06 with increasing fluorination (Fig. 3g and Supplementary

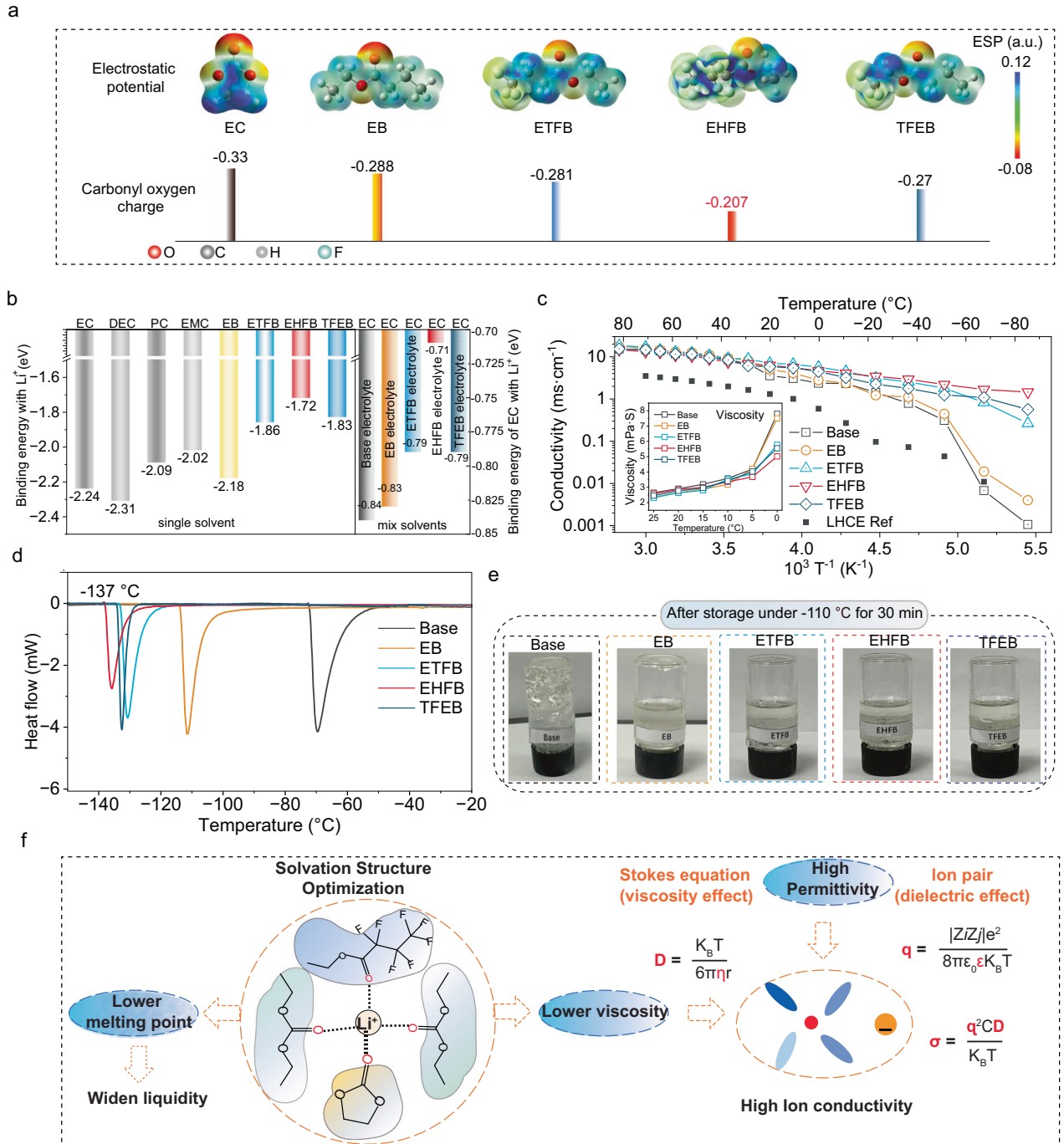

**Fig. 2 | The relationship between the charge distribution of solvent molecular and the physical properties of electrolyte. a** Electrostatic potential (ESP) maps showing the charge distributions of the solvent molecules considered in this work. **b** Binding energies of Li+ with different solvents. **c** Conductivity of the base electrolytes and the electrolytes containing EB, ETFB, EHFB, and TFEB. **d** DSC curves of the above electrolytes. **e** Digital images of different electrolytes after storage at −110 °C for 30 min. **f** Relationship of solvation structure and physical properties of electrolyte.

Fig. 8, Supplementary Table 6). This agreed well with the calculated energies for binding of Li+ with the solvents (Fig. 2b). Moreover, the coordination number of EHFB increased sharply from 0.06 at 25 °C to 0.56 at −70 °C (Fig. 3h). This indicated that EHFB becames more active in solvation of Li+ at LT. The low-ε solvent dominated the solvation structure together with the cosolvent, resulting in radical overall weaken interactions between Li+ and all solvents in the inner solvation sheath (Fig. 3j). These features facilitated Li+ desolvation and thus enhanced the electrochemical kinetics at LT.

Temperature-dependent FTIR and Raman spectral analyses further confirmed the contribution of the cosolvent to Li+ solvation along with the main solvents. (Fig. 3c–f, Supplementary Figs. 10–11). The IR peaks for free and solvated C = O (in the region of 1660-1870 cm$^{-1}$) were fitted with Voigt functions[27] (Fig. 3d, Supplementary Figs. 12–16). The peak assignments are presented in Supplementary Table 7-8. The ratios of solvated to free EC ($R_1$) and solvated EC to solvated DEC ($R_2$) (calculated with Eq. 1 and Eq. 2) were used to quantify the relative abundance of solvated EC and the competition

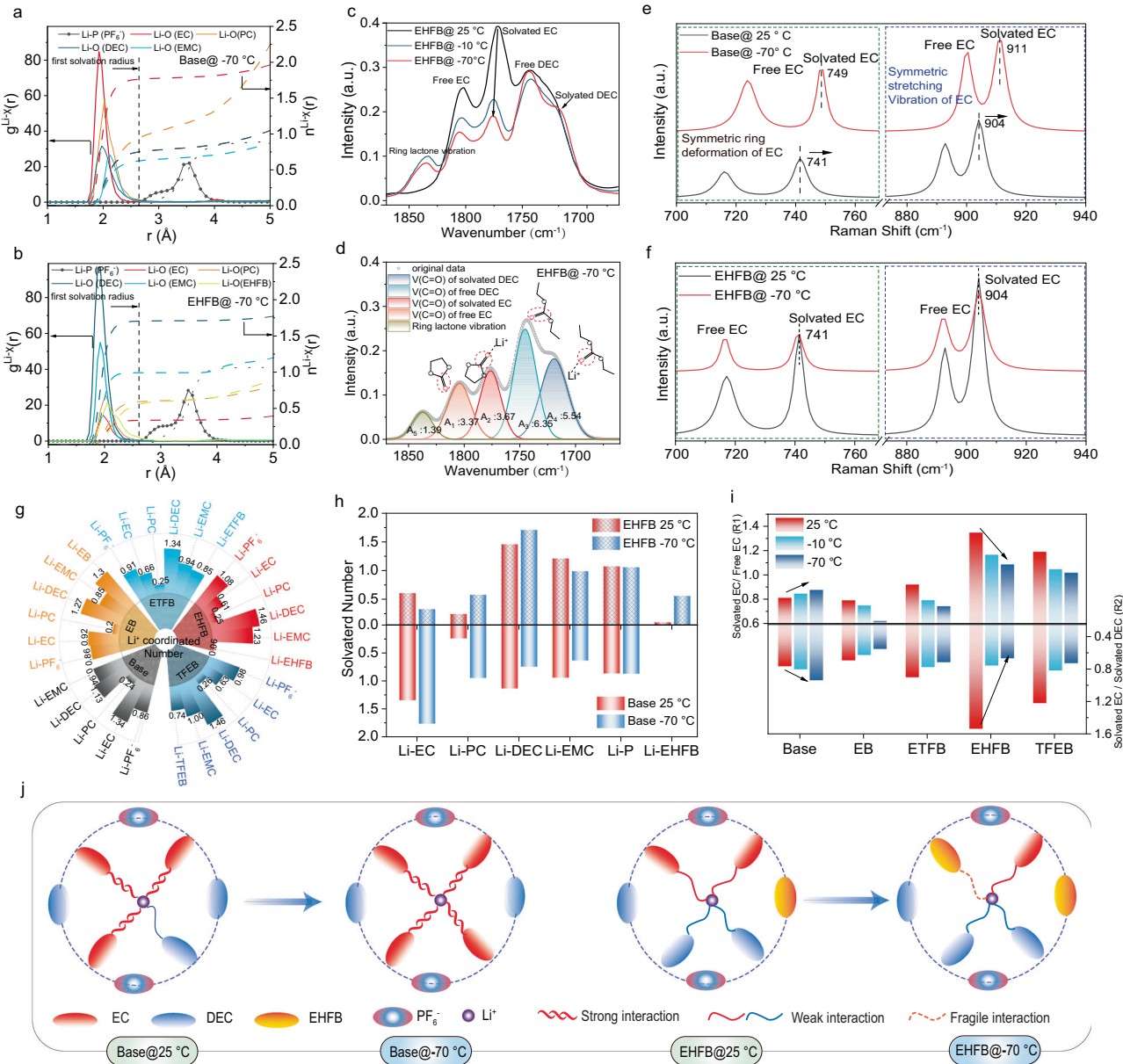

**Fig. 3 | Unlocking the solvation structure of electrolytes.** Calculated radial distribution functions (g(r)) and coordination numbers (n(r)) at −70 °C for base electrolyte (**a**) and EHFB electrolyte (**b**) based on molecular dynamics (MD) simulations. **c** The C = O regions of the FTIR spectra of EC and DEC in EHFB electrolyte at different temperatures. **d** Fitting spectra with Voigt function for EHFB electrolyte of −70 °C. Raman spectra for symmetric ring deformation of EC and stretching vibration of the (**C−O**) bond in base electrolyte (**e**) and EHFB electrolyte (**f**).

**g** Coordination numbers for the five studied electrolytes at 25 °C based on the calculated RDFs. **h** Coordination numbers of all solvent components with Li⁺ in the base and EHFB electrolytes at 25 °C and −70 °C. **i** Ratios of solvated EC to free EC, and solvated EC to solvated DEC in the base and EHFB electrolytes (derived from FTIR spectra) at different temperatures. **J** Illustration of solvation structure changes in the base and EHFB electrolytes upon lowering the temperature from 25 °C to −70 °C.

for coordination between EC and DEC[27].

$$R_1 = \frac{A_{solvated\,EC}}{A_{free\,EC}} \quad (1)$$

$$R_2 = \frac{A_{solvated\,EC}}{A_{solvated\,DEC}} \quad (2)$$

where $A_{solvated\,EC}$, $A_{free\,EC}$ and $A_{solvated\,DEC}$ are the integrated area intensities of the vibrational bands corresponding to the C = O groups of solvated EC, free EC, and solvated DEC, respectively (Fig. 3d). A quantitative analysis revealed that both the $R_1$ and $R_2$ values decreased as the temperature dropped from 25 °C to −70 °C (Fig. 3i) upon the

introduction of a cosolvent. For instance, the $R_1$ value for the EHFB electrolyte decreased from 1.34 (at 25 °C) to 1.09 (at −70 °C) along with a decrease in $R_2$ from 1.53 (at 25 °C) to 0.66 (at −70 °C) (Fig. 3i, Supplementary Table 8). This trend clearly illustrated that the cosolvent promoted coordination between the DEC and Li⁺, which was accompanied by weakened interactions between the EC and Li⁺, especially at LT. This translated into evolution of the first solvation shell structure from EC-dominant to DEC-dominant. Note that the $R_2$ value decreased faster for the EHFB electrolyte than for ETFB and TFEB (Fig. 3i, Supplementary Table 8), indicative of its greater selectivity for tailoring the DEC-dominant solvation structure at LT. Conversely, for the base electrolyte, both $R_1$ and $R_2$ increased as the temperature decreased from 25 °C to −70 °C, corroborating the EC-dominant

solvation over a wide temperature range (Fig. 3i). Furthermore, this change was accompanied by significant blueshifts of both solvated EC peaks in the Raman spectra (from 741 to 749 cm$^{-1}$ and from 904 to 911 cm$^{-1}$) with respect to the EHFB electrolyte (Fig. 3e, Supplementary Fig. 18), further indicating that the coordination interaction between Li$^+$ and EC was much stronger in the base electrolyte at LT[16,23]. This reduced its practical utility under cold conditions. These features, including the reduced R$_1$ and R$_2$ values (Fig. 3i) and suppressed blueshift of the C-O band (Fig. 3f), supported the hypothesis that the designed cosolvents unlocked the EC, thus tailoring a solvation structure with the DEC dominating and the cosolvent involved, and ultimately promoting a weak solvation structure (Fig. 3j). Together, the results highlight the significant advantage of our solvation design in weakening the interactions between Li$^+$ and the solvents in the electrolyte, thus facilitating Li$^+$ desolvation and improving the LT performance.

## Promotion of Li$^+$ desolvation and improvement of electrochemical kinetics

DFT calculations demonstrated that the fluorinated cosolvent dramatically lowered the Li$^+$ desolvation energy for the EC/DEC electrolyte (Fig. 4a), and this trend was highly fluorination dependent. The higher the fluorination degree of the cosolvent, the smaller the desolvation energy (from 1.1 eV for 4EC to 1.0 eV for EC-2DEC-EB, 0.93 eV for EC-2DEC-ETFB, and 0.83 eV for EC-2DEC-EHFB). This resulted from the strongly electron-withdrawing effects of the fluorine substituents, which enhanced the coordination between Li$^+$ and the low-ε solvents, and formed a DEC dominated, overall weakened solvation structure, thereby facilitating the desolvation process (Fig. 4b).

The impact of the fluorinated cosolvent on Li$^+$ desolvation was further supported by an analysis of the distribution of relaxation times (DRT) (see Supplementary Information for details) (Fig. 4c, d)[36–38]. This analysis classified different electrochemical processes by their local maxima in a continuous distribution function[39–41]. Both R$_{ct}$ and R$_{SEI}$ showed strong temperature dependence. R$_{ct}$ dominated at low temperature, while R$_{SEI}$ prevailed at high temperature (Supplementary Fig. 20), consistent with previous reports[38]. Specifically, the EHFB electrolyte had the lowest R$_{ct}$ of 1.0 Ω at −60 °C (Fig. 4d), followed by TFEB (3.2 Ω), and ETFB (4.7 Ω) (Supplementary Fig. 19–21). These values were much smaller than that of the base electrolyte (8.8 Ω) (Fig. 4c), highlighting the important role of cosolvents in facilitating charger transfer. This was also confirmed by the lower activation energies for charge transfer (Fig. 4f), and the activation energy for the LCO/Gr pouch cell with the EHFB electrolyte was estimated to be 10.9 kJ·mol$^{-1}$, approximately one third of that for the base electrolyte (30.4 kJ·mol$^{-1}$). The activation energies increased in the order EHFB < TFEB < ETFB, in good agreement with the trend for desolvation energies (Fig. 4a). It is worth noting that the sluggish kinetics at the graphite anode were reported to be the main challenge that lowered the potential for Li$^+$ intercalation into graphite to below 0 V (vs. Li/Li$^+$)[8,9,25,30], which was also evidenced by the much larger impedance of the graphite anode compared to the LCO cathode at LT (Supplementary Fig. 22). DRT mapping of Gr/Li half-cells over two electrochemical cycles at −10 °C in the base and EHFB electrolytes (Fig. 4g, h) revealed that the EHFB system exhibited a much smaller R$_{ct}$ than the base electrolyte, confirming that the introduction of a cosolvent facilitated charger transfer, especially in a graphite anode operating at low temperature. In addition, a significant decrease in R$_{SEI}$ was also identified (Fig. 4g, h) with a much lower activation energy for Li$^+$ transport through the SEI in full cells with EHFB (19.8 kJ·mol$^{-1}$) (Fig. 4f) compared with the base electrolyte (29.3 kJ·mol$^{-1}$) (Fig. 4e). Moreover, unlike the base case (Fig. 4g), the R$_{SEI}$ in the EHFB system varied slightly with much smaller values (Fig. 4h), suggesting that the SEI derived from the EHFB case was highly conductive and much more robust than that in the base electrolyte.

## Role of the solvation structure in determining SEI properties

The component/structural evolution of the SEI layer on the cycled graphite electrodes was characterized with time-of-flight secondary ion mass spectrometry (TOF-SIMS) (Supplementary Figs. 24–26 and Supplementary Fig. 31). The depth profiled TOF-SIMS data showed that organic moieties (CH$_2^-$, CO$_3^-$, C$_2$H$_3$O$^-$, and C$_2$H$_3$O$_2^-$) were mainly concentrated on the outer face of the SEI (Fig. 5c, d and Supplementary Fig. 28), while inorganic LiF species (LiF$_2^-$, Li$_2$F$_3^-$, Li$_3$F$_4^-$, Li$_4$F$_5^-$) and OH$^-$ were more prevalent at the inner side of the SEI (Fig. 5a, b, Supplementary Fig. 27), consistent with previous reports[42–44]. The assignments of the fragment ions and their potential sources are listed in Supplementary Table 9, including three main organic components: ROCO$_2$Li (representing CO$_3^-$ fragment), CH$_3$COLi (C$_2$H$_3$O$^-$) and CH$_3$COOLi (C$_2$H$_3$O$_2^-$), which were produced through electrochemical reduction of the EC, DEC and carboxylate, respectively. In the base case, the ROCO$_2$Li signal strengthened as the temperature was decreased from 25 °C to −10 °C (Fig. 5c and Supplementary Fig. 29a), while the CH$_3$COLi signal decreased (Supplementary Fig. 29b), and almost no CH$_3$COOLi signal was detectable (Fig. 5d and Supplementary Fig. 29c). This suggested that EC rather than DEC was the main solvent undergoing reduction in the base case at LT. Conversely, when cooling to −10 °C, the ROCO$_2$Li signal decreased and the CH$_3$COLi signal increased when using the EHFB electrolyte (Fig. 5c, Supplementary Fig. 29), implying enhanced DEC reduction and suppressed EC reduction at LT in the EHFB case. This was consistent with the previously discussed results (Fig. 3g–k) showing that the solvation of Li$^+$ by EC was promoted at LT in the base electrolyte but mitigated in the EHFB case. In addition, the detectable CH$_3$COOLi signal suggested that the EHFB cosolvent contributed to the solvation structure and participated in film formation on the graphite surface at LT (Fig. 5d), in agreement with the calculated LUMO energies (Supplementary Table 2). Furthermore, more LiF species were detected in the interface derived from the EHFB electrolyte (Fig. 5a, b), and their amounts increased as the temperature decreased (Supplementary Fig. 27c, d). The differences in SEI properties were reflected in the 3D spatial distribution overlay of organic SEI components (C$_2$H, ROCO$_2$Li) and inorganic species (LiF, Li$_2$O) at different temperatures (Fig. 5g). Specifically, the organic component (i.e., C$_2$H) was found only in the outermost layer of the SEI with low contents for the EHFB cases, and more LiF was present throughout the underlying layers with increasing amounts at LT. This may have resulted from more EHFB participating in solvation and contributing to SEI formation at LT, as proven in Fig. 3h. In contrast, the base electrolyte-derived SEI delivered a much smaller amount of LiF (Fig. 5a, Fig. 5d). These "friendly" anionic components are highly conductive interfacial species[42,45], and thus facilitate electrochemical kinetics, as discussed above.

Furthermore, in the case of the EHFB electrolyte, the dissolution and crossover of Co$^+$ were effectively suppressed (Fig. 5e, f and Supplementary Fig. 30a–d) under both RT and LT conditions. This was possibly due to oxidation resistance of the fluorinated electrolyte, with which the cathode structure was well protected from damage at high voltages of ~4.5 V, as verified by the LSV result (Supplementary Fig. 41). This highlighted the significant advantages of fluorinated cosolvents in high voltage applications and the excellent properties of the SEIs. In contrast, a large amount of Co$^+$ was dissolved and then crossed over to the graphite anode cycled with the base electrolyte, which is known to be highly destructive for the SEI[46] (Fig. 5e, f and Supplementary Fig. 30a–d). Together, these results explained the superior desolvation kinetics and LT performance observed for the EHFB electrolyte (Fig. 4c–h).

## Comparative electrochemical performance at low temperature

Coupling the EHFB electrolyte with commercial 1Ah NCM811/Gr and LCO/Gr pouch cells yielded excellent low temperature electrochemical performances. After 200 cycles at −10 °C with a cut-off potential of

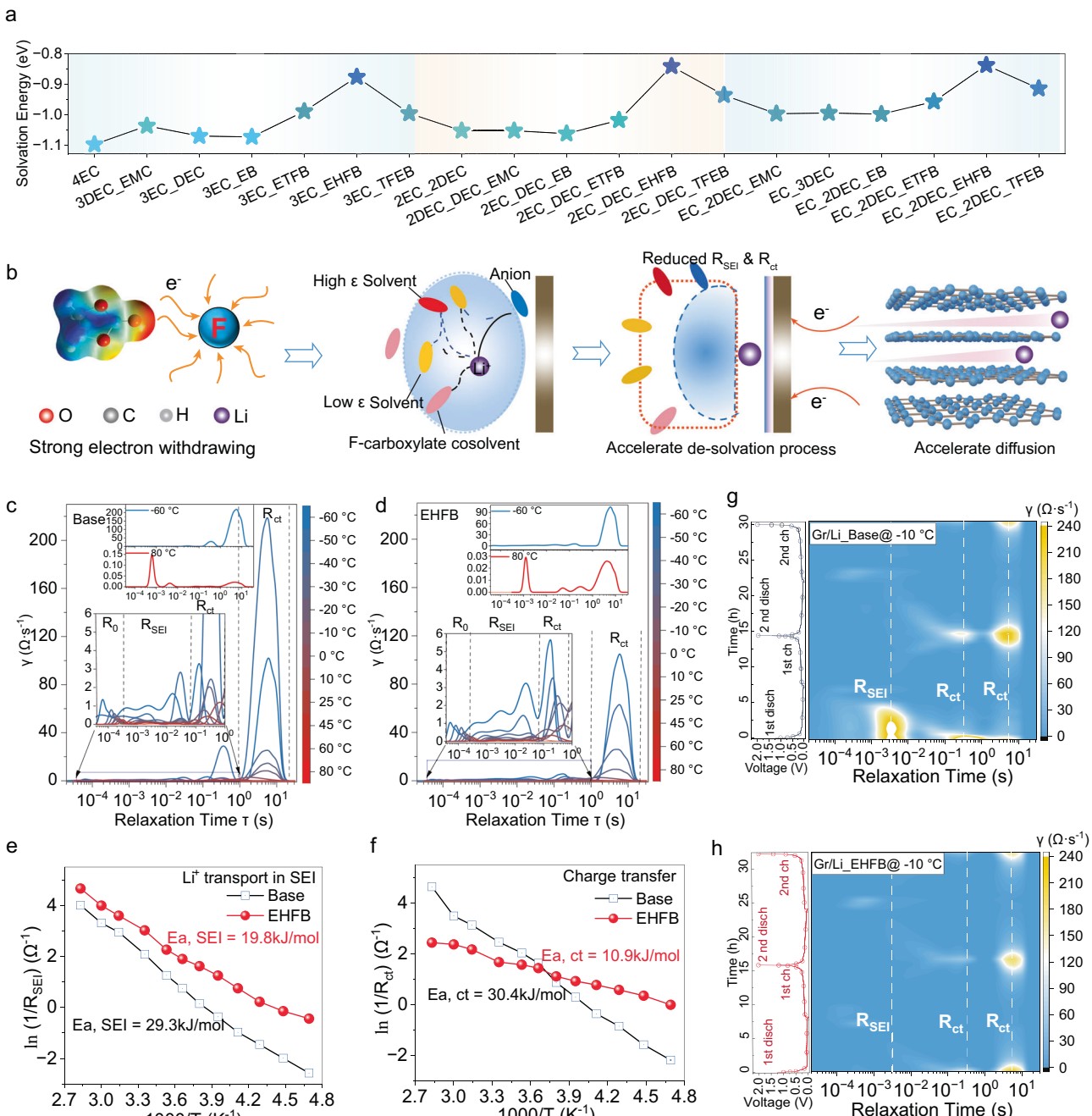

**Fig. 4 | Desolvation behaviour and its effect on electrochemical kinetics. a** Li⁺ solvation/desolvation energies in different electrolytes. **b** Illustration of fluorinated cosolvent's key role in weakening the interactions between Li⁺ and solvent molecules, accelerating desolvation and facilitating Li⁺ diffusion. Temperature-dependent distribution of relaxation times (DRT) plot derived from EIS data for the base electrolyte (**c**) and the EHFB electrolyte in LCO/Gr pouch cells (**d**). Corresponding activation energies derived by Arrhenius fitting for $R_{ct}$ (**e**) and $R_{SEI}$ (**f**) with the base and EHFB electrolytes. In-situ DRT data representing two cycles of graphite/Li half-cells with the base electrolyte (**g**) and the EHFB electrolyte (**h**) at −10 °C.

4.5 V, the capacity retention rates were 97.94% for NCM811/Gr and 86.70% for LCO/Gr (Fig. 6a, b) with an average coulomb efficiency of 99.6% and 99.5%, respectively (Supplementary Fig. 48). After being charged at −40 °C, the NCM811/Gr and LCO/Gr cells containing the EHFB electrolyte delivered discharge capacities of 576 mAh (50% of RT capacity) and 503 mAh (46% of RT capacity), respectively (Supplementary Figs. 43b–44b). Impressively, the NCM811/Gr cell with the EHFB electrolyte exhibited a discharge capacity of 364 mAh even after being charged at −60 °C. In stark contrast, cells with the base electrolyte nearly failed when charged below −30 °C (Supplementary Figs. 43a–44a). While achieving cycling below −60 °C remains challenging, we have made significant progress in enabling low-temperature discharge after full charging at room temperature. This approach is designed to simulate the practical operating conditions of electric vehicles in cold weather. Utilizing the RT charge- LT discharge protocol, the LCO/Gr pouch cell with the EHFB electrolyte retained a high capacity of 830 mAh at −40 °C, corresponding to 73.9% of their RT capacity. Notably, these cells still retained ~60% of their RT capacity even at −70 °C (Fig. 6d). Similar outstanding performance was also achieved with the NCM811/Gr pouch cells, which retained 61% of their RT capacity at −70 °C (Supplementary Fig. 36). The modified electrolyte enabled the cell to power an electric fan at ~−100 °C

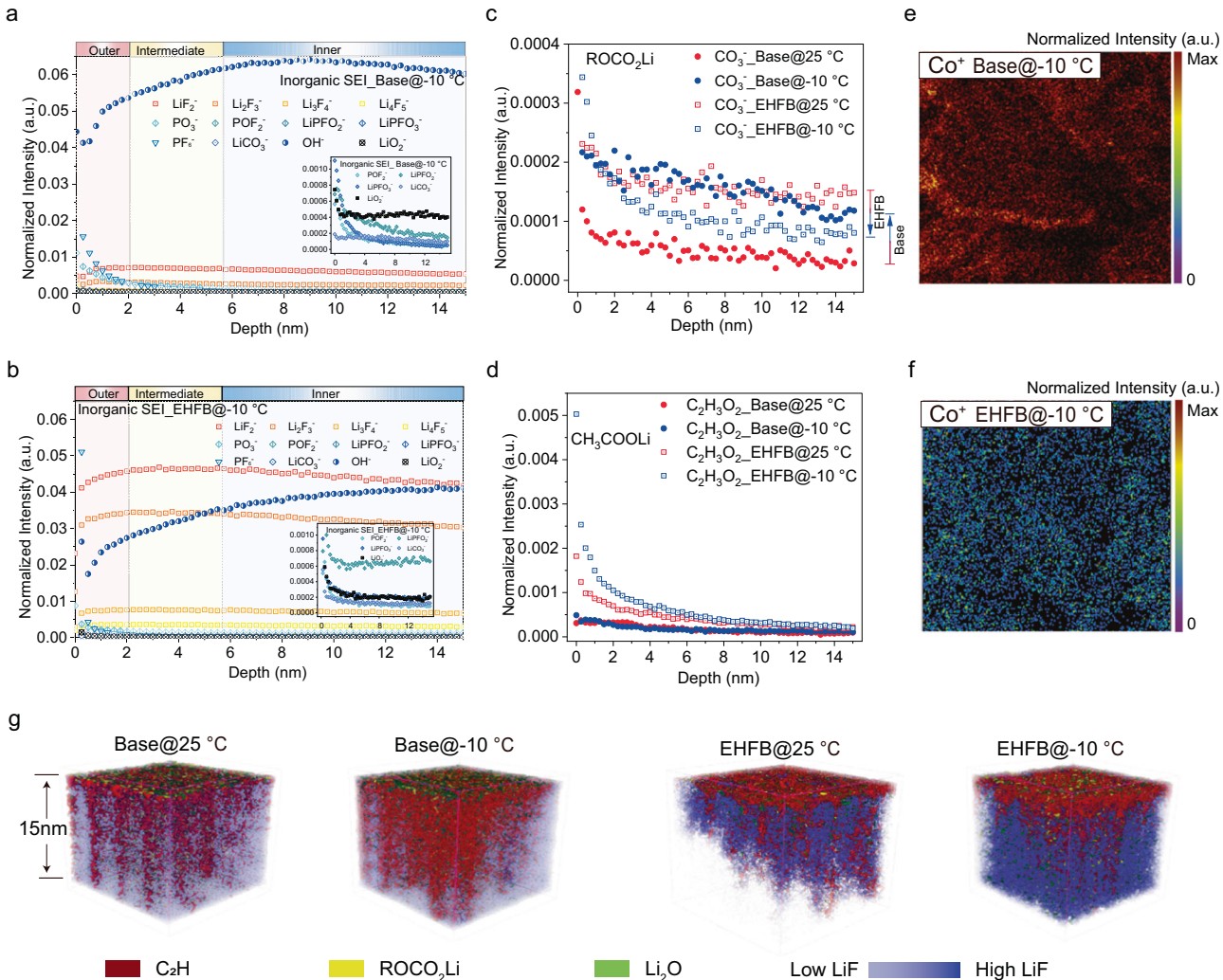

**Fig. 5 | TOF-SIMS characterization of the features and structural evolution of SEI at the graphite anode.** Depth profiles of functional groups in the inorganic SEI after prolonged cycling at −10 °C for the base electrolyte (**a**) and the EHFB electrolyte (**b**). Depth profiles of the ROCO$_2$Li component (**c**) and CH$_3$COOLi component (**d**) in the SEI formed from base and EHFB electrolytes at 25 °C and −10 °C. The spatial distribution of Co$^+$ on the graphite anode for the base electrolyte (**e**) and EHFB electrolyte (**f**). **g** 3D reconstructions of the topmost 15 nm of the graphite anode SEI formed at 25 °C and −10 °C for the base electrolyte and EHFB electrolyte cells.

(Supplementary Fig. 40, Supplementary Movie S1–S3) after being charged at room temperature, with the accuracy and stability of low temperatures confirmed in Supplementary Movie S3–S11. This demonstrated the viability of the new electrolyte at extremely low temperatures, since usable batteries were achieved at temperatures below the condensation point of CO$_2$ (−78 °C). In contrast, both LCO/Gr and NCM811/Gr cells using the base electrolyte exhibited poor performance, retaining only ~80% of their RT capacity at −10 °C and completely failing at −40 °C (Fig. 6c, Supplementary Fig. 36a). Furthermore, the pouch cells with the EHFB electrolyte (Supplementary Figs. 34–35) exhibited excellent cycling stability at both high (60 °C) and normal (25 °C) temperatures, delivering 800 mAh in the 55th cycle at 60 °C and negligible capacity loss over 100 cycles at 25 °C. These compelling findings demonstrated the feasibility and practicality of employing the proposed cosolvent for all temperature applications.

The low-temperature performance of a battery is limited by the graphite anode[8,9], which results from lithium dendrites that compromise the safety performance. This is proven by the fact that the graphite anode-based cells exhibited substantial capacity fading when cycled at −10 °C (Fig. 6a, b) compared to lithium anode-based cells with a base electrolyte (Supplementary Fig. 33). Herein, the Gr/Li cell was specifically evaluated at low temperature. Impressively, the Gr/Li cell with the EHFB electrolyte retained 57% and 95% of its RT charge capacity at −70 °C and −20 °C, respectively, after being discharged at room temperature. These values were much higher than those with the base electrolyte (20% and 69% RT capacity, respectively, Fig. 6e). Moreover, the Gr/Li cell utilizing the EHFB electrolyte still delivered a discharge specific capacity of 95 mAh·g$^{-1}$ even during discharge and charge at −50 °C (Supplementary Fig. 45b), this constituted smaller polarization than the cell with the base electrolyte, which failed at −30 °C (Supplementary Fig. 45a). The polarization difference increased dramatically with decreasing temperature, with ~1.4 V for the base case and only 0.48 V for the EHFB case at −50 °C (Supplementary Fig. 45f). These results emphasize the crucial role of the EHFB cosolvent in enhancing the low-temperature properties of the graphite anode. In addition, the Gr/Li cells with fluorinated cosolvents, especially EHFB, showed remarkably stable cycling performance at −10 °C, retaining 99% of their capacity over 120 cycles (Supplementary Fig. 37c). This reinforced the practicality of using graphite-based cells in cold conditions with the designed electrolyte. Conversely, the cells containing the base electrolyte suffered from dramatic capacity losses in the first 20 cycles (from 249.3 to 28.5 mAh·g$^{-1}$), followed by gradual increases in capacity before ultimately stabilizing at 100 mAh·g$^{-1}$ after

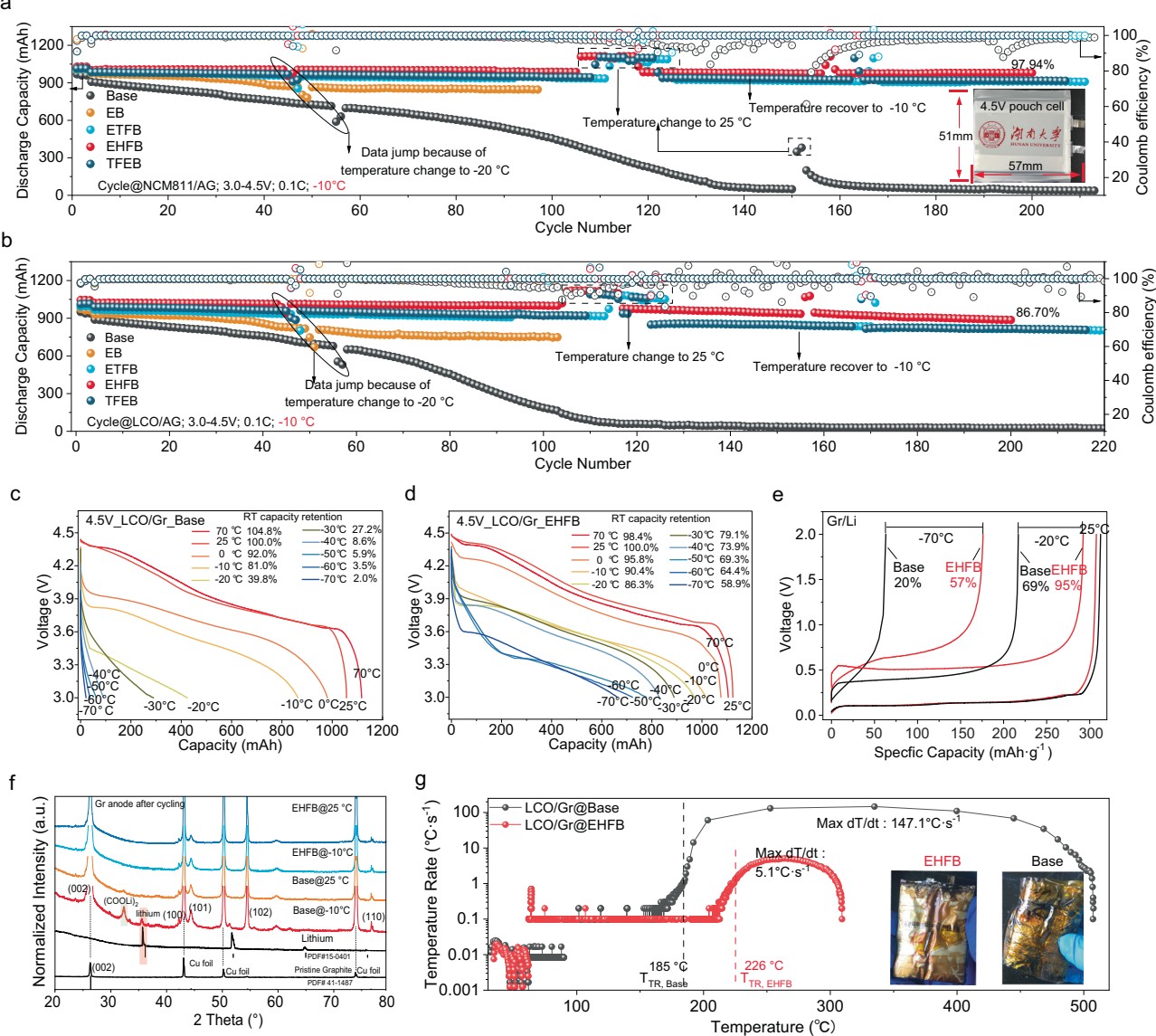

**Fig. 6 | Low temperature performance of the commercial LiCoO₂/Graphite (LCO/Gr) and LiNi₀.₈Co₀.₁Mn₀.₁O₂/Graphite (NCM811/Gr) pouch cell with the designed electrolytes and common electrolyte. a**, **b** Cycling behaviour of NCM811/Gr cell (**a**) and LCO/Gr (**b**) with different electrolyte under −10 °C, the optical photos of pouch cell is inserted in Fig. 6a. **c**, **d** The temperature-dependent discharge profiles of LCO/Gr pouch cells with Base electrolyte (**c**) and EHFB electrolyte (**d**) with RT charge- LT discharge protocol. **e** Charge profiles of Gr/Li cells with different electrolyte at different temperatures. **f** XRD pattern of the dismantled graphite after the long cycling at −10 °C and 25 °C. **g** Temperature dependence of temperature rate of the fully charged LCO/Gr cells via ARC test. The insets show the photos after the test.

120 cycles (Supplementary Fig. 37c). This abnormally poor cycling behaviour was attributed to insufficient lithiation at LT, which resulted from the accumulation of lithium dendrites. Moreover, the Gr/Li cells with the EHFB electrolyte exhibited exceptional rate performance (Supplementary Fig. 37a), delivering 77.7 mAh·g⁻¹ at 5 C and promptly recovering to their initial capacity of 331.6 mAh·g⁻¹ at 0.2 C. This remarkable rate capability suggested potential applicability in fast charging applications with the designed cosolvent, considering the common issue of lithium dendrites formation with graphite anode operation at low temperature and high C-rate conditions[10]. This concerned lithium dendrite issue on the graphite surface under LT[47] was extensively analyzed via interfacial and structural characterization and electrochemical characterization. The XRD and XPS analyses provided direct evidence of lithium dendrite formation in the base electrolyte. A distinct lithium peak at 35.8° was observed in the XRD pattern of the graphite anode cycled at −10 °C in the base electrolyte (Fig. 6f), which was absent in the EHFB case. The XPS spectra also confirmed the

presence of metallic lithium (-52.2 eV)[48–51] on the graphite surface cycled in the base electrolyte, whereas it was undetectable in the EHFB case (Supplementary Fig. 38). TOF-SIMS imaging (Supplementary Fig. 23) and SEM images (Supplementary Fig. 39) also revealed the morphology of lithium dendrites on the graphite anode. Moreover, the differential capacity profiles (dV/dQ) of cells using the base electrolyte showed much more prominent Li plating peaks compared with those with the cosolvent after cycling at −10 °C (Supplementary Fig. 42)[47]. All of these findings corroborated that lithium dendrite formation was effectively suppressed in the modified batteries at low temperatures, demonstrating the great superiority of our electrolyte design.

The concerned safety performance was assessed with accelerating rate calorimetry (ARC) (Fig. 6g and Supplementary Fig. 46). The results illustrated a significant improvement in the safety of the LCO/Gr pouch cell with the EHFB electrolyte compared with the base electrolyte. In particular, the EHFB electrolyte resulted in a substantially reduced max dT/dt of only 5.1 °C s⁻¹ and a much higher

thermal runaway temperature ($T_{TR}$, 226 °C), indicating a more pronounced tendency to avoid thermal runaway incidents. Moreover, the much lower maximum temperature was reduced to 309 °C, signifying a substantial reduction in the total energy released during the thermal runaway process[52–56]. These findings underscored the benefits of our electrolyte design in safe performance.

## Discussion

We have demonstrated a solvation design strategy with traditional EC-based electrolytes to improve the low-temperature performance of LIBs. By introducing fluorinated cosolvents, we weakened the strong coordination of EC to Li⁺ and achieved a fluorine-dependent transition from an EC-dominated solvation structure to a DEC-dominated solvation structure, especially at low temperatures. This design facilitated the desolvation of Li⁺ while maintaining the high dielectric property of EC. Moreover, the fluorinated cosolvents also contributed to the formation of a fluorine-rich SEI that enhanced the stability of the graphite anode under cold conditions. As a result, we have achieved remarkable improvements in low-temperature properties, such as a wider liquidity range (retaining liquid at −110 °C), better conductivity (1.46 mS·cm⁻¹ at −90 °C), a more facile desolvation process, and suppressed lithium dendrite growth. These enhancements allowed the 1 Ah 4.5 V graphite-based pouch cells to be cycled stably over 200 cycles at −10 °C with only a 2% capacity loss and a retained 334 mAh capacity when charged and discharged at −60 °C for one cycle. Furthermore, the cell was still discharged at −70 °C with 60% of its room temperature capacity and power electrical devices at extremely low temperatures of ~ −100 °C after being charged at 25 °C. This work involving a solvation design strategy with traditional EC-based electrolytes provides a unique approach to developing lithium-ion batteries suitable for use in extreme environments.

## Methods

### Electrolyte preparation and battery fabrication

The base electrolyte was prepared by dissolving 1 M LiPF$_6$ in a mixture of carbonate solvents (EC:PC:DEC:EMC = 2:1:3:4 by volume). The test electrolytes were prepared with the same concentration of LiPF$_6$ but with half of the DEC replaced with a fluorinated cosolvent or EB, giving a solvent composition of EC:PC:DEC:EMC:cosolvent = 2:1:1.5:4:1.5 by volume. Four cosolvents were tested: EB and its fluorinated analogues ETFB, EHFB, and TFEB.

One Ah dry pouch cells manufactured by Li-Fun Technology which have 8 cathode layers (with 5.9 g for LCO and 5.6 g for NCM811 active material) and 9 anode layers (with 3.94 g and 4.46 g, respectively), were injected with 4.3 g of electrolyte for NCM811/Gr cells or 2.1 g for LCO/Gr cells in a glovebox filled with argon. The standard cell formation process was then completed by performing gas release and reseeding to obtain cells with electrochemical windows of 3–4.5 V. The specific capacity is equal to the capacity divided by the mass of active material, and the actual general capacities delivered by the LCO/Gr pouch cell and NCM811/Gr pouch cell are both 1100 mAh, that is, specific capacity for NCM811/Gr cell is 1100 mAh/5.6 g = 196.4 mAh·g⁻¹, for LCO/Gr cell is 1100 mAh/5.9 g = 186.4 mAh·g⁻¹. 1 C refers to the current required to fully charge the battery to the cut-off potential after 1 h. For the pouch cell with 1Ah, 1 C is 1 A. And for the coin cells, 1 C in LCO/Li and NCM811/Li is the current density related to the nominal specific capacity of 178 mAh·g⁻¹ and 188 mAh·g⁻¹, so that the 1 C is 178 mA·g⁻¹ and 188 mA·g⁻¹, respectively.

NCM811/Li, LCO/Li and Gr/Li coin cells were made with a LiNi$_{0.8}$Mn$_{0.1}$Co$_{0.1}$O$_2$ cathode, LiCoO$_2$ cathode and graphite anode, respectively (all sourced from Guangdong Canrd New Energy Technology Ltd.), with specific surface areas of 1.88 m² g⁻¹, 0.24 m² g⁻¹, and 5.49 m² g⁻¹, respectively (measured via nitrogen adsorption-desorption isotherm (JW-BK200C, Beijing JWGB SCI and TECH)).

### Electrochemical measurements

The charge/discharge behaviours of the batteries (coin cells and pouch cells) were tested with a battery test system (CT-4008T-5V6A, CT-4008T-5V10mA, Neware, Shenzhen, China, and CT2001A, Wuhan Land Electronics Co. Ltd). The batteries to be tested were placed in the climatic chamber (GMC-71, Espec, Guangzhou, China) and brought to an appropriate onset temperature (−20 °C, −10 °C, 25 °C, 60 °C) for the cycling test. The main low temperature test was conducted with RT charge- LT discharge mode at −70 °C, −60 °C, −50 °C, −40 °C, −30 °C, −20 °C, −10 °C, 0 °C, 25 °C, 70 °C, and the LT charge-discharge mode was also employed to evaluate the electrolyte's performance under low-temperature charging performance. Cycling test and cyclic voltammetry (CV) test were performed with the electrochemical window of 3.0 V-4.5 V, and linear sweep voltammetry (LSV) test was carried out within OCV-7 V in a three-electrode configuration, with platinum plate as the work electrode and lithium foil as the counter electrode and reference electrode. Electrical impedance spectroscopy was performed in the climatic chamber with a temperature range from −60 °C to 80 °C with the frequency range of 0.01 Hz to 100 kHz using an electrochemical workstation (Solartron 1455 A, Solartron Group, UK) with an amplitude of 10 mV.

### Electrolyte characteristic

The conductivity of the electrolytes were determined with a conductivity meter (Mettler Toledo, Shanghai, China) in a cold trap containing a mixture of ethyl alcohol and liquid nitrogen, and the temperature was adjusted by varying the ratio of the two substances. FTIR spectra were acquired with a Fourier transform tnfrared spectrometer (Nicolet, iS50 FT-IR, Thermo Scientific, KBr tablet, wave number 4000-600 cm⁻¹). Raman spectra were acquired with a WITEC alpha 300 R Raman system (532 nm laser, laser power 2 mW). The electrolyte was stored at the onset temperature before the FTIR and Raman tests, and the temperature controlling system was that used for the conductivity tests.

### Material characterizations

XRD measurements were performed with a Rigaku Miniflex X-ray diffractometer using a Cu Kα$_1$ source to investigate the lithium phases of the cycled graphite anodes, which were wrapped in Teflon film in a vacuum. An X-ray photoelectron spectroscopy system (XPS, Ulvac-Phi, PHI Versaprobe 4) equipped with a vacuum transfer vessel accessory (KW-ST lab, www.kewei-scitech.com) was used to analyse the lithium metal on the surface of the cycled graphite anode. XPS were calibrated by setting the binding energy for the hydrocarbon (C–C/C–H) in C 1 s spectra to 285 eV. The lithium dendrite morphology of the disassembled graphite anode was observed with a field emission scanning electron microscopy (FE-SEM, Jeol, JSM-7610FPlus) with a vacuum transfer box (Navi Innovation Co. Ltd, navi-sci.cn).

### TOF-SIMS

During TOF-SIMS depth profiling, a combination of mass spectrometric analysis and sputter ion source etching was employed to collect depth profile curves of the SEI components from the surface to the interior. The mass spectrometry analysis used a 30 keV pulsed ion beam of Bi$_3^{++}$ as the ionising source, with a DC beam current of 9.7 nA and a spectrum acquisition area of 150 μm × 150 μm, over a mass range of 2-500 amu. The etching process involved a 3 keV Ar⁺ ion beam for sample removal, with a DC beam current of 100 nA and an etching area of 600 μm × 600 μm. TOF-SIMS depth profiling offered high depth resolution. Throughout the depth profiling process, the depth of secondary ion information obtained during the mass spectrometric analysis was -1–2 molecular layers, and the depth per cycle for sputter etching was -0.25 nm (a rate calculated based on a SiO$_2$ standard sample). Mass scale calibration was performed with common fragment

ions (positive mode: $CH_3^+$, m/z:15.02; $C_2H_3^+$, m/z:27.02; $C_3H_5^+$, m/z:41.04; negative mode: $CH^-$, m/z:13.00; $C_2H^-$, m/z:25.00; $C_4H^-$, m/z:49.00). Data processing was performed with PHI TOF-DR software (Physical Electronics, Minnesota, USA).

The graphite electrode preparing procedure before mass spectrometry is presented in detail below.

Disassembly of the pouch cells: After undergoing long-term cycling at both room temperature and low temperature, the pouch cells were disassembled within a glove box with a controlled atmosphere of $H_2O/O_2 \leq 0.01$ ppm.

Negative electrode treatment: The negative electrode (graphite electrode) was carefully separated from the disassembled pouch cells. Subsequently, the electrode was thoroughly washed with dimethyl carbonate (DMC) to remove any residual electrolyte or contaminants. After the washing process, the electrode was dried and then sealed in an aluminium-plastic bag.

Vacuum drying: The sealed aluminium-plastic bag containing the graphite electrode was placed in a vacuum drying oven set at 60 °C for 12 h. This step ensured complete removal of any remaining solvent or moisture from the electrode surface.

Transfer to glove box: After the vacuum drying process, the samples were transferred (while ensuring that they were not exposed to air) to transfer vessels located inside the glove box, where mass spectrometry was performed.

## Computational methods
Geometry optimization, energy calculations, and electronic structure analyses of electrolyte and solvent molecules were performed with the Gaussian 16 package with the B3LYP functional[57] and the 6−311 G(d,p) double-zeta basis set[58]. Frequency analyses were performed with the same basis set to verify the stabilities of the optimized structures. The solvation effects for the complexes of $Li^+$ with different molecules were evaluated with the SMD implicit solvation model with acetone ($\varepsilon = 20.49$) as the continuum solvent. Molecular dynamics simulations of the common electrolyte (EC:PC:DEC:EMC = 20:10:30:40) and the four cosolvent-containing electrolytes (EC:PC:DEC:EMC:cosolvent = 20:10:15:40:15 by volume) were performed with the Forcite module in Material Studio software with the COMPASS III ab initio forcefield[59] and the Nosé thermostat[60]. Before constructing the solvation models, all the involved molecules were optimized using Dmol3 module[61] with BLYP functional and spin-unrestricted settings in a fine quality. During this process, the ESP charges of the atoms were calculated. The numbers of different molecules in the five systems are shown in Supplementary Table 5. Before performing simulations with the canonical ensemble (NVT) at the specified temperature for 10 ns, the density of each system was equilibrated with the NPT ensemble for tens of picoseconds. The dynamic calculations were performed with a fine quality and a time step of 1 fs, and initiated with the current charges and random velocities. The equilibrated dimensions and densities of the solvation model units are listed in Supplementary Table 10.

## Reporting summary
Further information on research design is available in the Nature Portfolio Reporting Summary linked to this article.

## Data availability
The data that support the plots within this paper and other findings of this study are available from the corresponding author upon request. Source data are provided with this paper.

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

## Acknowledgements

This work was financially supported by the National Natural Science Foundation of China (U21A2081, 22075074), National Key Research and Development Program of China (2022YFE0206300), Outstanding Young Scientists Research Funds from Hunan Province (2020JJ2004), Major Science and Technology Program of Hunan Province (2020WK2013).

## Author contributions

J.L.L. and Y.Q.C. conceived the project. J.L.L., X.M.H., L.D.X., B.H.L. and X.L.F supervised the project. Y.Q.C. carried out most of the experiments and wrote the manuscript. Y.Z., W.Z., P.T.X., P.G. and N.T. contributed to part of the preparation and characterization of the electrolyte and the electrode. Q.H. conducted the calculations. J.T. provided all pouch cells. All authors discussed and commented on the manuscript.

## Competing interests

The authors declare no competing interests.
