## [Peer Review File · Nature Communications]

Breaking Solvation Dominance of Ethylene-Carbonate via Molecular Charge Engineering Enables Battery Operation at Lower TemperatureREVIEWER COMMENTS

Reviewer #1 (Remarks to the Author):

The manuscript is interesting, however all the claims are not very supportive with collected battery data.

1) Fig. S36I Temperature dependent discharge profiles of NCM811/Gr pouch cells. a, Base, b, EHFB. We need to see battery discharging and charging curves for all the temperatures.

2) Fig. S40I Photos of an electric fan powered by the 4.5V LCO/Gr pouch cell using EHFB electrolyte at superior low temperature -98.4°C. This is very crude way to demonstrate low temperature cycling. Please show charging capability at that temp.

3) Fig. 6 shows only Gr/Li charging, full cell charging and discharging need to be presented with showing polarization.

4) Title claims "Breaking Solvation Dominance of EC via Electron Engineering Enables Battery Operation Below -100°C" in fact its too much hyped and need to reframe.

5) These cells also retain ~60% of their room-temperature discharge capacity at 21 -70 °C, and miraculously remain functional even below -100 °C. Please provide long C-D cycling data.

6) After modifying the electrolyte, what happens to thermal safety. Is the full cell battery going to be safer or more prone to thermal runaway. ARC (accelerated rate calorimetry) on pouch cells will prove that.

7) This design strategy of breaking the solvation dominance of EC via electron engineering can be extended to other alkali-metal-ion batteries at extremely low temperature. Why authors believes that? Sodium and potassium systems are very different. Please measure and prove it, else remove such vague statements.

Reviewer #2 (Remarks to the Author):

I have now reviewed the manuscript, titled 'Breaking Solvation Dominance of EC via Electron Engineering Enables Battery Operation Below -100°C'. This manuscript addresses an important topic of maintaining high performance of lithium ion batteries at subzero temperatures. The authors have conducted quite a bit of work – it is actually a comprehensive manuscript suitable for a journal like Journal of power sources. Given the significance of the results, I can see why the authors are interested in publishing in communications with most of the data moved to supporting info. There are some missing information that needs to be discussed to make the manuscript worthy of publications.

1) Line 101/102 makes the claim that increasing the degree of fluorination of ethyl butyrate decreases Li⁺ ion interaction with solvent. Do authors mean with itself (ethyl butyrate) or other solvent molecules? Because after so many Fluorine attachments, dipole simply shifts within the molecule and would still strongly interact with Li⁺ ion.

- 2) Figure 1c says “ether side fluorination” there is no ether present, only an ester
- 3) Lines 165 through 186 discuss how the coordination number of EC is decreased and the coordination number of DEC is increased, upon using EHFB cosolvent. This effect is even more pronounced at low temperatures. However, if the goal is to create weaker solvent Li⁺ ion interactions, how is the benefit of reducing the EC coordination number not diminished by the increase in coordination number of DEC?
- 4) The method to sputter to a precision of just a few nanometers for depth profiling needs to be fully described
- 5) How were the graphite samples prepared for mass spectrometry analysis? The methods section does not give enough detail about this.
- 6) Normalized discharge capacity of the cells was not explicitly mentioned, but doing some basic math seems to show near full theoretical capacity of LCO (274 mAh/g) based off Figure 6a and having used 1Ah pouch cells. Most cells perform at about 50-60% of their theoretical capacity at even room temperature, so how are these cells able to retain near perfect discharge capacity?
- 7) Concerning Figure 6g, how is metallic lithium visible to the naked eye in the optical images provided? Why is XRD or XPS data not given to verify this statement? If analyzed via SEM, a special detector is needed in order to detect lithium, which is not mentioned in the paper or supplemental information.
>>>This is a serious weakness for a manuscript that contains so much studies.
- 8) Line 412 states that cells retained 800 mAh/g at cycle 55 at 60°C, yet none of the materials of LCO, NMC or graphite have capacities nearly that high. Can this attribute be explained?

Prof. Dr. Jilei (J. L.) Liu
College of Materials Science and Engineering
Hunan University
Changsha 410082, China
E-mail: liujilei@hnu.edu.cn
Phone: +86-731-88664009
Fax: +86-731-88664010

Point-to-Point Responses:

Reviewer #1:

The manuscript is interesting, however all the claims are not very supportive with collected battery data.

Our response: We deeply appreciate the invaluable feedback provided by the reviewer, and we sincerely apologize for any confusion that may have arisen from part of inappropriate claims. As suggested, we have included more data (Fig S43-46) and discussions in main text (page 16-20), and made corresponding corrections (Page 2, 6, 17-21), to highlight the novelty of this work. More details can be found below.

Comment 1

Fig. S36| Temperature dependent discharge profiles of NCM811/Gr pouch cells. a, Base. b, EHFB. We need to see battery discharging and charging curves for all the temperatures.

Our response: As requested, we have included the comprehensive all-temperature charge-discharge curves of the NCM811/Gr pouch cell (Fig S43). And the results are detailed discussed in the main text (Page 16-17, line 338-344), as described below.

The NCM811/Gr cell utilizing EHFB electrolyte demonstrates significantly enhanced charge and discharge capacities at each temperature, compared to the cell using base electrolyte which failed at -40°C, with remarkable room temperature (RT) capacity retentions of 50% and 46% after being charged at -40°C and -50°C, respectively. Notably, the NCM811/Gr cell with EHFB electrolyte is capable of delivering discharge capacity of 364 mAh (32% RT capacity) even after being charged at -60°C (Figure R1).

Figure R1. a-b, temperature dependent charge and discharge profiles of 4.5V NCM811/Gr pouch cell using base (a) and EHFB electrolyte (b) under low temperature charge-discharge protocol at 0.05C. c-d, the corresponding charge and discharge capacity and coulombic efficiency. e-f, the capacity retention of room temperature at all temperatures for discharge (e) and charge (f).

*Note: The charge capacity showed slightly increased at -50 °C is due to a rest at room temperature for 1 hour before cycle at -50°C, to relieve the alarm of long-term low temperature operation of the test chamber.

Comment 2

Fig. S40| Photos of an electric fan powered by the 4.5V LCO/Gr pouch cell using EHFB electrolyte at superior low temperature -98.4°C. This is very crude way to demonstrate low temperature cycling. Please show charging capability at that temp.

Our response: Thanks for the comments. To avoid any further confusion, the photos images were removed, and the charging behavior of the 4.5V LCO/Gr pouch cell at different temperatures are provided in Fig. S44, with detailed discussion in the main text (Page 16-17, line 338-347).

Although the modified electrolyte reported here is still failed in powering the cell to cycle under extreme temperature conditions as low as -98.4°C, it can be well discharged at -98.4°C after being fully charged at room temperature. The RT charge-LT discharge protocol is also widely applied in the field of low temperature battery research (*Nature Energy*, 2019, 4(10): 882-890; *Advanced Materials*, 2023, 35(3): 2208340, *Advanced Energy Materials*, 2022, 12(48): 2201801; *Chinese Physics Letters*, 2023, 40(8): 86101), given that this mode is strategically aligned with the envisioned real-world applications of electric vehicles during cold weather- charging at RT and running at LT.

In this work, a breakthrough has also been made in terms of enabling the cell to be cycled below -40 °C with the EHFB electrolyte, with 46% and 37% RT capacity retention when cycled at -40 °C and -50 °C, respectively (Figure R2), while the common cell failed cycling at -30 °C.

Figure R2. a-b, temperature dependent charge and discharge profiles of 4.5V LCO/Gr pouch cell using base (a) and EHFB electrolyte (b) under low temperature charge-discharge protocol at 0.05C. c-d, the corresponding charge and discharge capacity and coulombic efficiency. e-f, the capacity retention of room temperature at all temperatures for discharge (e) and charge (f).

Comment 3

Fig. 6 shows only Gr/Li charging, full cell charging and discharging need to be presented with showing polarization.

Our response: As suggested, the charging and discharging profile of the Gr/Li cell under various low temperatures has been provided in Fig. S45, along with detailed discussions in the main text (Page 18-19 line 381-387).

The Gr/Li cell utilizing EHFB electrolyte still delivers remarkable results, showcasing a discharge specific capacity of 95 mAh/g even at -50°C, while that with base electrolyte failed at -30°C (Figure R3). Furthermore, the EHFB electrolyte significantly lowers polarization in the cycled Gr/Li cells compared to the base electrolyte. The polarization difference increases dramatically with decreasing temperature. Specifically, the polarization was ~1.4V at -50°C for the base case, while the EHFB case exhibited only 0.48V. These findings underscore the clear advantages of EHFB electrolyte, especially in addressing the issue toward the more concerned graphite anode at low temperature.

Figure R3. a-b, temperature dependent charge and discharge profiles of Gr/Li half cells using base (a) and EHFB electrolyte (b) under low temperature charge-discharge protocol at 0.05C. c-d, the corresponding charge and discharge capacity and coulombic efficiency. e, the capacity retention of room temperature at all temperatures. f, the corresponding temperature dependent polarization voltage of Gr/Li cell.

Comment 4

Title claims "Breaking Solvation Dominance of EC via Electron Engineering Enables Battery Operation Below -100°C" in fact its too much hyped and need to reframe

Our response: We highly value the insightful comments provided by the reviewer. In response, we have made significant improvements to the title, now rephrased as '*Breaking Solvation Dominance of EC via Molecular Charge Engineering Enables Battery Discharge at -100°C*'.

Comment 5

These cells also retain ~60% of their room-temperature discharge capacity at -70 °C, and miraculously remain functional even below -100 °C. Please provide long C-D cycling data.

Our response: Responses to comments 1-3 showed that the modified cell faced cycling challenges below -60°C, and we apologize for the difficulty in providing long-term cycling data below -20°C, as the long C-D cycling declined sharply at -20°C (Fig 6a-b).

Nevertheless, we have achieved significant progress in resolving the low temperature issues, with 98% capacity retention after 200 long C-D cycles at -10°C (Fig 6a), and enabling ~60% RT capacity retention at -70°C, and being able to release energy in extreme environments of ~-100°C after being fully charged at room temperature (Fig 6d, Fig S40 and Movie S1). These results are superior over

previous reports (Fig 6c, Fig S43a-S44a), demonstrating the greatly practical feasibility of our electrolyte design.

Expectantly, we are actively addressing this issue of extreme low-temperature charging in our ongoing research, and are committed to achieving long-term charge-discharge cycling of batteries at even lower temperatures.

Comment 6

After modifying the electrolyte, what happens to thermal safety. Is the full cell battery going to be safer or more prone to thermal runaway. ARC (accelerated rate calorimetry) on pouch cells will prove that.

Our response: As suggested, we have conducted an Accelerating Rate Calorimetry (ARC) test, as detailed in Fig S46 and Fig 6g (Page 20, line 416-425).

The results of this test serve to illustrate a significant enhancement in safety when utilizing our EHFB electrolyte, compared to the base electrolyte. Specifically, the EHFB electrolyte led to a substantially reduced max dT/dt of only $5.1\text{ }^{\circ}\text{C}\cdot\text{s}^{-1}$, and much higher T_{TR} ($226\text{ }^{\circ}\text{C}$) (Figure R4). These findings indicate a more pronounced tendency to avoid thermal runaway incidents. Additionally, the much lower maximum temperature ($309\text{ }^{\circ}\text{C}$), signifies a substantial mitigation of the total energy released during the thermal runaway process (*Joule*, 2020, 4(4): 743-770; *Advanced Materials*, 2022, 34(2): 2106335; *Advanced Energy Materials*, 2022, 12(44): 2201964). Overall, these results further highlight the safety advantages of our electrolyte design.

Figure R4. Thermal safety evaluation of the 1Ah LCO/Gr pouch cell based on accelerating rate calorimetry (ARC). a-b) Temperature/ voltage profiles of the cell using Base electrolyte (a) and EHFB electrolyte (b). c) The explanation of the key point temperature during ARC test. d) Temperature dependence of temperature rate of the fully charged cells. The insets show the photos after the test.

Comment 7

This design strategy of breaking the solvation dominance of EC via electron engineering can be extended to other alkali-metal-ion batteries at extremely low temperature. Why authors believes that? Sodium and potassium systems are very different. Please measure and prove it, else remove such vague statements.

Our response: We sincerely apologize for confusion that may have arisen regarding the inappropriate claims. Indeed, significant differences are existing among Li, Na, K ion batteries. The specific electrolyte formula can not be directly transferable. The statement has been therefore corrected as ‘**This strategy of disrupting solvation dominance of EC through molecule charge engineering, opens new avenues for advanced electrolyte design toward better lithium-ion batteries.**’ (Page 2, line 21-23)

Reviewer #2:

Comment 1

Line 101/102 makes the claim that increasing the degree of fluorination of ethyl butyrate decreases Li⁺ ion interaction with solvent. Do authors mean with itself (ethyl butyrate) or other solvent molecules? Because after so many Fluorine attachments, dipole simply shifts within the molecule and would still strongly interact with Li⁺ ion.

Our response: Thanks for your invaluable comments. The comprehensive experimental results (Fig 3) in combination with detailed theoretical simulations (Fig 2a-b) reveal that the fluorination of ethyl acetate induces two notable effects: 1) Weakening the interaction between Li⁺ and fluorinated-ethyl acetate solvent itself, and 2) weakening the coordination of Li⁺ with surrounding solvent molecules. The former is evidenced by the decreased electronegativity of the carbonyl oxygen after fluorination (Fig. 2a) and the reduced binding energy of Li⁺ with a single solvent after fluorination (Fig. 2b), as well as the decreased coordination number of fluorinated-ethyl acetate solvents (Fig. 3g). The latter one is supported by the reduction in the coordination number of EC after fluorination (Fig. 3e, f, g, h, i) and the reduced binding energy of Li⁺ with EC in a mixed solvent system after fluorination (Fig. 2b). Moreover, a higher degree of fluorination results in a more pronounced weakening effect.

The underlying mechanism is well illustrated in Fig R5. It is true that the introduction of fluorine atoms primarily induces a shift in the dipole moment within the molecule itself. However, the changes in charge distribution due to dipole modification also impact the electronegativity of the molecule and the surrounding molecules. Specifically, the electronic cloud of the fluorinated butyl acetate solvent molecules congregates at the fluorinated methylene end (-CF₃), while the electronic cloud of the surrounding solvent molecules concentrates on the carbonyl oxygen (C=O). As a result, the dipole-dipole interaction between fluorinated ethyl butyrate and the surrounding EC molecule changes after fluorination, ultimately leading to a weak solvation of Li⁺. This phenomenon is explained by “dipole-dipole effect” (*ACS Energy Lett.* 2022, 7, 3545–3556; *Energ. Environ. Sci.* 2023, 16(2): 546-556).

Correspondingly, the discussion has been supplemented in the main text (Page 6, line 119-125) and Fig. S47.

How to change the coordination of surrounding solvent molecules after fluorination

Figure R5. The effect of fluorinated solvents on the dipole-dipole interactions among surrounding solvent molecules and their coordination with Li^+ .

Comment 2

Figure 1c says “ether side fluorination” there is no ether present, only an ester

Our response: In adherence to the nomenclature rules of organic chemistry, we have revised the names of the fluorinated groups as “propyl side fluorination” and “ethoxy side fluorination”, and corresponding modifications are also made throughout the whole text. (Page 5, line98)

Figure R6. Chemical structure of the selected ethyl butyrate (EB) and its fluorinated analogue cosolvents.

Comment 3

Lines 165 through 186 discuss how the coordination number of EC is decreased and the coordination number of DEC is increased, upon using EHFB cosolvent. This effect is even more pronounced at low temperatures. However, if the goal is to create weaker solvent Li^+ ion interactions, how is the benefit of reducing the EC coordination number not diminished by the increase in coordination number of DEC?

Our response: Thanks for the insightful comments. We would like to clarify that the weakening of ethylene carbonate (EC) coordination is accompanied by an increase in diethyl carbonate (DEC) coordination. In other words, the decrease in EC coordination invariably leads to a corresponding increase in DEC coordination. This phenomenon arises because of a constant total number of solvent molecules coordinating around Li^+ .

Clearly, the advantage of reducing EC coordination is not overshadowed by the increase in DEC coordination. This is attributed to the intrinsically low polarity nature of DEC with respect to EC. On one hand, since EC is a high polar solvent, thus a decrease in the coordination number of EC no doubt results in a weakening of solvation. On the other hand, an increase in the coordination number of DEC also leads to a weakening of solvation, given its low polarity. These two factors are causally related, and their combined effect ultimately promotes an overall weak solvation structure.

Moreover, the lower temperature, the more DEC coordination, the weaker solvent- Li^+ interaction, and the better low temperature performance. This highlights the novelty of this EHFB electrolyte design. (Page 10, line 202-204; Page 4, line 66-67)

Comment 4

The method to sputter to a precision of just a few nanometers for depth profiling needs to be fully described

Our response: As suggested, detailed description of the method have been supplemented in the experiment section of TOF-SIMS. (Page 23, line 498-508).

Comment 5

How were the graphite samples prepared for mass spectrometry analysis? The methods section does not give enough detail about this.

Our response: As suggested, more details about graphite anode preparing before the mass spectrometry test have been supplemented in the experiment section (Page 23-24, line 513-529).

“Disassembly of Pouch Cells: The pouch cells, after undergoing long-term cycling at both room temperature and low temperature, were disassembled within a glove box environment with a controlled atmosphere of $\text{H}_2\text{O}\backslash\text{O}_2\leq 0.01\text{ppm}$.

Negative Electrode Treatment: The negative electrode (graphite electrode) was carefully separated from the disassembled pouch cells. Subsequently, the electrode was thoroughly washed with dimethyl carbonate (DMC) to remove any residual electrolyte or contaminants. After the washing process, the electrode was dried and then sealed in an aluminum-plastic bag.

Vacuum Drying: The sealed aluminum-plastic bag containing the graphite electrode was placed in a vacuum drying oven set at 60°C for 12 hours. This step ensured complete removal of any remaining solvent or moisture from the electrode surface.

Transfer to Glove Box: Following the vacuum drying process, the samples were transferred (ensuring they were not exposed to air) to transfer vessels located inside the glove box, where the mass spectrometry testing was performed.”

Comment 6

Normalized discharge capacity of the cells was not explicitly mentioned, but doing some basic math seems to show near full theoretical capacity of LCO (274 mAh/g) based off Figure 6a and having used 1Ah pouch cells. Most cells perform at about 50-60% of their theoretical capacity at even room temperature, so how are these cells able to retain near perfect discharge capacity?

Our response: We apologize for confusion caused by the specific capacity calculation. we have now included detailed parameter information regarding the pouch cells in the experimental methods section of the manuscript (Page 21, line 455-457). This will facilitate data conversion and provide readers with a clear understanding of the specific capacity calculations for both LCO/Gr and NCM811/Gr pouch cells.

In this work, we utilized 1Ah LCO/Gr and NCM811/Gr pouch cells, with the total mass of active material in the cathode being 5.9g and 5.6g, respectively. The actual general capacities delivered by the LCO/Gr pouch cell and NCM811/Gr pouch cell are both 1100 mAh (Fig 6c-d, Fig S36a-b). The specific capacity is calculated by dividing the actual capacity by the mass of the active material in the cathode. For the LCO/Gr pouch cell, the specific capacity is $1100 \text{ mAh} / 5.9 \text{ g} = 186.4 \text{ mAh/g}$, achieving approximately 68% of its theoretical specific capacity (274 mAh/g). Similarly, for the NCM811/Gr pouch cell, the specific capacity is $1100 \text{ mAh} / 5.6 \text{ g} = 196.4 \text{ mAh/g}$.

Comment 7

Concerning Figure 6g, how is metallic lithium visible to the naked eye in the optical images provided? Why is XRD or XPS data not given to verify this statement? If analyzed via SEM, a special detector is needed in order to detect lithium, which is not mentioned in the paper or supplemental information. This is a serious weakness for a manuscript that contains so much studies.

Our response: We are sorry for the mistakes, the yellow color observed in the optical photo corresponds to LiC_6 . To make correction, the XRD and XPS analyses are provided to directly evident lithium dendrite formation, and the optical photo is replaced with XRD results in Figure 6f. An obvious lithium peak located at 35.8° was observed in the graphite electrode cycled at -10°C in the base electrolyte (Figure R7), which is absent in the EHFB electrolyte. This observation is further supported by the supplemented XPS analysis (Figure R8). The distinct peak corresponding to metallic lithium ($\sim 52.2 \text{ eV}$) (*Chem. Mater.* 2021, 33, 859–867, *Journal of The Electrochemical Society*, 1994, 141(9): 2379) is observed in the base electrolyte, whereas it was undetectable in EHFB case. All these findings, corroborate that lithium dendrite is effectively suppressed in the modified batteries at low temperatures, demonstrating the great superiority of our electrolyte design. The discussion has been supplemented in the main text (Page 19-20, line 401-415) and Fig S38.

Figure R7. XRD pattern of the graphite anode cycled in both electrolytes.

Figure R8. X-ray photoelectron spectroscopy (XPS) profiles of Li 1s of the dismantled graphite anode after the long cycling for both electrolytes at -10°C and 25°C .

Regarding the SEM measurement, it is true that lithium can not be detected without the special detector. In this work, we use SEM with a vacuum transfer box to observe the morphology features of dendrites (Fig S39) on the surface of graphite, rather than detecting metallic lithium directly. This observation via FE-SEM is widely reported (*Science*, 2018;359(6383):1513-1516; *Proceedings of the National Academy of Sciences*, 2017, 114(46): 12138-12143). The corresponding description has been supplemented in the experiment section (Page 22-23, line 493-495).

Comment 8

Line 412 states that cells retained 800 mAh/g at cycle 55th at 60°C, yet none of the materials of LCO, NMC or graphite have capacities nearly that high. Can this attribute be explained?

Our response: We sincerely apologize for the confusion caused by erroneously using the specific capacity unit, mAh/g, instead of the correct capacity unit, mAh. We have now rectified this error in the main text, ensuring the proper usage of the capacity unit, mAh, for all relevant data and calculations. (Page 17, 361)

REVIEWERS' COMMENTS

Reviewer #1 (Remarks to the Author):

The revision is carried out. The claims are slightly modified. However, following things still needs to be modified before the final acceptance of this manuscript.

The revised title "Breaking Solvation Dominance of EC via Molecular Charge Engineering Enables Battery Discharge at -100°C" still needs to be further modified as follows.

"Breaking Solvation Dominance of EC via Molecular Charge Engineering Enables Battery Operations at Lower Temperature".

This is because the battery is charged at room temperature and discharged at -100C. Being the battery rechargeable, it has to have promise to charge and discharge at the same (-100c) temperature.

The following added new statement is also misleading.

Furthermore, the cell can still discharge at -70 °C with 60% of their room temperature capacity, and power electrical devices at extremely low temperatures of ~ -100 °C after being charged at 25 °C, which meets the practical needs of envisioned real-world applications that electric vehicles run at LT but charge at RT in cold weather.

The charging of electric vehicles mostly happens outside at that cold/low temperature. Not everyone has heated car garages. Many EV catches fires when the circulating ethylene glycol does not maintain room temperature during charging. Thus, real world application of low temperature batteries are for space applications. e.g. on the surface of Moon or Mars the temperature will be as below as -140C. WITHOUT external heating or cooling (that we currently do) batteries shall be charging and discharging at that low temperature.

As previously mentioned the method of lowering down the battery temperature is very crude and may not be accurate based on "the conductivity of the electrolytes was determined with a conductivity meter (Mettler Toledo, Shanghai, China) in a cold trap containing a mixture of ethyl alcohol and liquid nitrogen whose temperature was adjusted by varying the ratio of the two substances. The liquid nitrogen freezes at -196C attenuation is required in liquid-gas phase in a specially designed system for accurate measurements.

Reviewer #2 (Remarks to the Author):

The authors have addressed almost all of my comments. Some of my comments were addressed only in the responses - they should be included in the manuscript so other readers also understand. Specifically, if the authors include the response to the following issue in the manuscript then it should be acceptable for publication:

Lines 165 through 186 discuss how the coordination number of EC is decreased and the coordination number of DEC is increased, upon using EHFB cosolvent. This effect is even more pronounced at low temperatures. However, if the goal is to create weaker solvent Li⁺ ion interactions, how is

the benefit of reducing the EC coordination number not diminished by the increase in coordination number of DEC?

Prof. Dr. Jilei (J. L.) Liu
College of Materials Science and Engineering
Hunan University
Changsha 410082, China
E-mail: liujilei@hnu.edu.cn
Phone: +86-731-88664009
Fax: +86-731-88664010

Point-to-Point Responses:

Reviewer #1:

The revision is carried out. The claims are slightly modified. However, following things still needs to be modified before the final acceptance of this manuscript.

Comment 1

The revised title "Breaking Solvation Dominance of EC via Molecular Charge Engineering Enables Battery Discharge at -100 °C" still needs to be further modified as follows. "Breaking Solvation Dominance of EC via Molecular Charge Engineering Enables Battery Operations at Lower Temperature". This is because the battery is charged at room temperature and discharged at -100 °C. Being the battery rechargeable, it has to have promise to charge and discharge at the same (-100 °C) temperature.

Our response: Thanks for the commons. As suggested, we have revised title as “Breaking Solvation Dominance of Ethylene-Carbonate via Molecular Charge Engineering Enables Battery Operations at Lower Temperature”.

Comment 2

The following added new statement is also misleading. “Furthermore, the cell can still discharge at -70 °C with 60% of their room temperature capacity, and power electrical devices at extremely low temperatures of ~ -100 °C after being charged at 25 °C, which meets the practical needs of envisioned real-world applications that electric vehicles run at LT but charge at RT in cold weather”. The charging of electric vehicles mostly happens outside at that cold/low temperature. Not everyone has heated car garages. Many EV catches fires when the circulating ethylene glycol does not maintain room temperature during charging. Thus, real world application of low temperature batteries are for space applications. e.g. on the surface of Moon or Mars the temperature will be as below as -140C. WITHOUT external heating or cooling (that we currently do) batteries shall be charging and discharging at that low temperature.

Our response: Thanks for the comments. To avoid any further misunderstanding, the statement has been revised as follows “Furthermore, the cell can still discharge at -70 °C with 60% of its room

temperature capacity, and power electrical devices at extremely low temperatures of ~ -100 °C after being charged at 25 °C". (Line 438-441, Page 16)

Comment 3

As previously mentioned the method of lowering down the battery temperature is very crude and may not be accurate based on "the conductivity of the electrolytes was determined with a conductivity meter (Mettler Toledo, Shanghai, China) in a cold trap containing a mixture of ethyl alcohol and liquid nitrogen whose temperature was adjusted by varying the ratio of the two substances. The liquid nitrogen freezes at -196C attenuation is required in liquid-gas phase in a specially designed system for accurate measurements.

Our response: Thanks for the constructive feedback, as suggested, we have advanced the cold trap device to ensure the temperature accuracy, durability and reliability from two aspects:

i) **Double insulation structure design ensured durability of low temperature:** We incorporated an insulating layer onto the exterior of the container, which held a mixture of liquid nitrogen and ethanol. The container's top was sealed with circular thermal insulation foam. Subsequently, the entire container was placed within a cylindrical foam insulation box surrounded by ice cubes to further minimize heat release (as depicted in Figure R1 and illustrated in Figure R2).

ii) **Mutual calibration with glass thermometer and digital thermometer ensured accurate temperature control.** The glass thermometer was calibrated using a digital thermometer to confirm the accuracy of measuring temperatures. The temperature fluctuations at -100 °C remained within ± 2 °C for 11.5 minutes. Notably, the temperature held between -94 °C and -93 °C last for an impressive 37 minutes, as depicted in Figure R3-R4 and Supplementary Movie 3-4. Consequently, the temperature rise rate (0.14 °C·min⁻¹) within the ultra-low temperature zone ensures the precise, stable and reliable low-temperature testing, including low-temperature conductivity testing and electric fan operation experiments at ~ -100 °C. Similar approaches are also demonstrated in previous reports including *Nature Energy*, 2019, 4(10): 882-890; and *Energy Environ. Sci.*, 2023,16, 1024-1034 etc, validating the practical feasibility.

Figure R1. The physical diagram of the cold trap device for low temperature test.

Figure R2. Schematic of the cold trap device for low temperature test.

Figure R3. Temperature monitoring during low-temperature test by the double check of digital thermometer and glass thermometer. The corresponding process was recorded as shown in Supplementary Movie 3-11.

Figure R4. Variation of temperature vs. time during the low temperature test.

Reviewer #2:

The authors have addressed almost all of my comments. Some of my comments were addressed only in the responses - they should be included in the manuscript so other readers also understand. Specifically, if the authors include the response to the following issue in the manuscript then it should be acceptable for publication:

Lines 165 through 186 discuss how the coordination number of EC is decreased and the coordination number of DEC is increased, upon using EHFB cosolvent. This effect is even more pronounced at low temperatures. However, if the goal is to create weaker solvent Li⁺ ion interactions, how is the benefit of reducing the EC coordination number not diminished by the increase in coordination number of DEC?

Our response: Thanks for reminding. As suggested, we have included all the corresponding responses into the manuscript toward the commons of the first review.

1. To address the concern of “how is the benefit of reducing the EC coordination number not diminished by the increase in coordination number of DEC?”, we add response in the manuscript as follows:

“The decrease in EC coordination inevitably led to a corresponding increase in DEC coordination considering a constant total number of solvent molecules coordinating each Li⁺. Clearly, the advantage of reducing EC coordination will not be overshadowed by an increase in DEC coordination, which is attributed to the intrinsically low polarity nature of DEC with respect to EC. On one hand, since EC is a high polar solvent, thus a decrease in the coordination number of EC no doubt results in a weakening of solvation. On the other hand, an increase in the coordination number of DEC also leads to a weakening of solvation given its low polarity. Therefore, these two factors synergistically promote the formation of an overall weaken solvation structure.” (Line 188-197, Page 7, marked with red colour)

2. To explain the decrease in Li⁺ ion interaction with both ethyl butyrate itself and main solvent molecules after fluorine attachments, we response in the manuscript as follows:

“These results demonstrated that fluorination not only reduced the coordination of ethyl butyrate itself but also weakened the interactions between the surrounding EC and Li⁺, and the latter can be explained by the “dipole-dipole effect” (**Supplementary Fig.47**). Specifically, the strong electron affinity of fluorine led to a shift in the charge distribution from C=O end to -CF₃ end, which changed the dipole-dipole interaction between the fluorinated ethyl butyrate and the surrounding EC molecule and eventually resulted in a weak solvation of Li⁺ by the EC.” (Line 133-140, Page 5, marked with red colour)

3. The XRD and XPS data has been provided and discussed to evident the lithium dendrite formation in the main text as follows:

“The XRD and XPS analyses provided direct evidence of lithium dendrite formation in the base electrolyte. A distinct lithium peak at 35.8° was observed in the XRD pattern of the graphite anode cycled at -10°C in the base electrolyte (**Fig. 6f**), which was absent in the EHFB case. The XPS spectra also confirmed the presence of metallic lithium ($\sim 52.2\text{ eV}$) on the graphite surface cycled in the base electrolyte, whereas it was undetectable in the EHFB case (**Supplementary Fig.38**).”
(Line 400-406, Page 14-15, marked with red colour)

4. The other response about the detail of TOF-SIMS test and the calculations of the normalized specific discharge capacity of pouch cells have been provided in the method section. (Line 432-436, Page 17 and line 480-511, Page 18-19)